# Pediatric Early Warning System (PEWS) Association with ICU Mortality in Children with Acute Lymphoblastic Leukemia: A Cohort Study from Kazakhstan

**DOI:** 10.3390/medicina61112054

**Published:** 2025-11-18

**Authors:** Yedil Kurakbayev, Abay Kussainov, Kuanysh Umbetov, Yernur Zikiriya, Yergali Sarsekbayev, Botagoz Turdaliyeva, Nazira Nurgozhayeva, Arai Tolemisova

**Affiliations:** 1Department of Anesthesiology, Resuscitation and Intensive Care, JSC “Scientific Center of Pediatrics and Pediatric Surgery”, 146 Al-Farabi Avenue, Almaty 050040, Kazakhstan; 2Department of Biostatistics and Epidemiology, Kazakhstan’s Medical University “KSPH”, Almaty 050060, Kazakhstan; zikiriaernur@gmail.com; 3Department of Science, JSC “Scientific Center of Pediatrics and Pediatric Surgery”, 146 Al-Farabi Avenue, Almaty 050040, Kazakhstan; info@pediatria.kz; 4Department of Transfusion, JSC “Scientific Center of Pediatrics and Pediatric Surgery”, 146 Al-Farabi Avenue, Almaty 050040, Kazakhstan; umbetovkuanysh92@gmail.com; 5Scientific Center of Urology Named After Academician B.U. Dzharbusynov, Almaty 050060, Kazakhstan; 6Department of Hematopoietic Stem Cell Transplantation, JSC “Scientific Center of Pediatrics and Pediatric Surgery”, 146 Al-Farabi Avenue, Almaty 050040, Kazakhstan; ergali1907@gmail.com; 7Department of Scientific Activity, Kazakh Scientific Center of Dermatology and Infectious Diseases, Raiymbek Avenue 60, Almaty 050002, Kazakhstan; bot.turd@gmail.com; 8Department of Anesthesiology and Intensive Care of Cardiac Surgery, JSC “Scientific Center of Pediatrics and Pediatric Surgery”, 146 Al-Farabi Avenue, Almaty 050040, Kazakhstan; nazira.nur131510@gmail.com; 9LLP “International Research Institute for Postgraduate Education”, Toraigyrov Street 49/1, Almaty 050043, Kazakhstan

**Keywords:** Pediatric Early Warning System, PEWS, acute lymphoblastic leukemia, ICU mortality, pediatric oncology, Kazakhstan, critical care, sepsis

## Abstract

*Background and Objectives*: Childhood acute lymphoblastic leukemia (ALL) carries substantial morbidity, mortality, and economic burden, particularly in middle-income countries. The Pediatric Early Warning System (PEWS) is designed to trigger timely escalation of care, yet its independent impact on survival among critically ill leukemic children has not been well defined in Kazakhstan and Central Asia. *Materials and Methods*: We conducted a retrospective review all ICU admissions for patients aged 0–18 years with ALL at the National Center of Pediatrics, Almaty, across two periods: pre-implementation (January 2020–December 2022) and post-implementation of 24 h PEWS monitoring (September 2023–December 2024). The primary outcome was ICU mortality. Seven domains of covariates—demographic, clinical history, transfusion, vital signs, symptoms, laboratory, and instrumental data—were extracted. Univariable and multivariable logistic regression models were used to assess associations with mortality. *Results*: Among 255 admissions (105 during PEWS implementation; 150 prior to PEWS implementation), overall ICU mortality was 21.7%. After adjustment, PEWS implementation was not associated with reduced ICU mortality (AOR 0.89), despite a lower unadjusted mortality (15.9% vs. 26.6%). The most clinically relevant secondary findings included strong associations between mortality and bilateral pneumonia (AOR 7.45), ≥4 episodes of hyperthermia within 24 h of ICU admission (AOR 5.42), and systemic inflammatory response syndrome (AOR 4.61). *Conclusions*: These findings suggest that, within this high-acuity cohort, inflammatory and cardiorespiratory derangements outweigh any potential survival benefit from ward-based PEWS surveillance. Optimizing outcomes will require integrating early warning systems with timely deterioration management, focused cardiopulmonary support, and resource allocation tailored to the clinical context—rather than relying solely on surveillance scores.

## 1. Introduction

Acute lymphoblastic leukemia (ALL) is the most common hematologic malignancy in children, with incidence highest in those younger than five years [1]. Worldwide, new cases of childhood ALL increased by 59% between 1990 and 2021, reaching approximately 169,000 annually in this age group [2]. During the same period, ALL-related deaths and disability-adjusted life years (DALYs) declined by about two-thirds [2]. Despite this progress, five-year survival still varies markedly by region. In a low-income Mexican cohort, five-year overall survival was only 31% [3], while investigators in Pakistan reported a rate of 52.9% [4]. By contrast, U.S. registry data for 2004–2009 report a five-year survival of 89% [5]. These disparities reflect the uneven distribution of diagnostic and therapeutic advancements, which have largely benefited high- and upper-middle–Socio-Demographic Index (SDI) settings.

In Kazakhstan, the incidence of childhood ALL reached 3.4 per 100,000, and all-cause mortality from pediatric oncohematological malignancies was 1.5 per 100,000 in 2021 [6]. Treatment protocols for ALL in Kazakhstan follow national clinical guidelines and typically include a prephase, induction and consolidation chemotherapy, central nervous system prophylaxis, and hematopoietic stem cell transplantation [7]. Therapeutic decisions are stratified based on immunophenotypic classification and cytogenetic risk factors [7].

While standardized treatment protocols have significantly improved survival rates for pediatric ALL, approximately one-third of patients still require intensive care unit (ICU) admission due to acute complications [8,9]. These complications commonly include sepsis, respiratory failure, tumor lysis syndrome, and neurological deterioration [10,11]. Such acute episodes often occur rapidly and unpredictably, highlighting the importance of early identification tools that are simple, scalable, and adaptable to varying resource levels.

In this context, the Pediatric Early Warning Score (PEWS) has emerged as a valuable bedside tool for the early detection of clinical deterioration [12]. PEWS integrates age-adjusted vital signs and simple clinical observations to identify children at risk of impending decompensation using a “track and trigger” system [12]. Originally developed for early identification of sepsis in high-resource pediatric settings, PEWS is now used as a part of early-warning surveillance and rapid-response activation strategies in high-, middle- and low-income countries [13,14]. Currently, its application has expanded beyond sepsis recognition to encompass broader indicators of clinical instability and physiologic deterioration [15].

Local implementation of PEWS at the National Center of Pediatrics in Almaty began in 2023, supported by staff training and standardized escalation protocols [16]. The tool is currently used to signal ICU transfer rather than diagnose infection per se. Pediatric ICUs in Kazakhstan often serve mixed patient populations, including medical, surgical, and oncohematological cases. However, a 2018 assessment identified the absence of a universal triage protocol in Kazakhstani ICUs, contributing to inefficiencies in recognizing critical deterioration and allocating beds [17]. While subsequent health reforms increased financing and updated care models [18], the impact on ICU triage practices remains unclear. In this setting, tools like PEWS may help bridge the gap between clinical deterioration and timely ICU admission [19]—especially in oncology patients who deteriorate rapidly and nonspecifically.

Existing studies focus largely on heterogeneous ward populations; few adjust for disease-specific risk factors such as relapse status, transfusion burden, or laboratory markers of organ dysfunction [20]. Consequently, it is unclear whether PEWS implementation offers incremental prognostic value once children with ALL deteriorate to the point of ICU admission.

This study aims to determine whether PEWS use is associated with ICU mortality among pediatric patients. Specifically, we quantify that association after systematically adjusting for patient demographics, clinical history and disease characteristics, transfusion history, vital signs, clinical symptoms, laboratory parameters, and instrumental test results. To our knowledge, this is the first study from Kazakhstan to explore PEWS utility in pediatric oncology, offering novel insights into its role in a mixed-resource ICU environment. By addressing this gap, we aim to inform resource-sensitive triage strategies for pediatric oncology in Kazakhstan and comparable settings.

## 2. Methods

### 2.1. Research Design and Study Population

We performed a retrospective review of hospital records for all patients aged 0–18 years diagnosed with acute lymphoblastic leukemia (ALL) who were admitted to the ICU for onco-hematological patients at the National Center of Pediatrics and Pediatric Surgery, Almaty, Kazakhstan. Two non-overlapping time frames were analyzed. Period 1 (January 2020–December 2022): PEWS monitoring had not yet been implemented; therefore, every ICU admission (from onco-hematological wards and emergency department) of children with ALL was included. Period 2 (September 2023–December 2024): after full implementation of PEWS monitoring in the four specialized oncohematology wards, only patients transferred from these wards were eligible. The temporal gap between the two periods allowed for staff training and consistent 24-h PEWS use before hospitalization. For patients with multiple ICU stays during a single hospitalization, only data from the final ICU admission were analyzed. For patients with several hospitalizations, the last hospitalization was used. Post-ICU mortality outcomes were not available. Inclusion criteria: Children aged 0–18 years with a confirmed diagnosis of ALL who were admitted to the ICU during either study period. Exclusion criteria: In Period 2, ICU admissions originating from the emergency department (*n* = 14) were excluded due to the lack of PEWS monitoring in that setting. The patient selection process is outlined in Figure 1.

### 2.2. Sample Size and Power

Given the retrospective nature of this study and the total available cohort size (*n* = 295), a formal a priori power calculation was not conducted. However, the sample size was maximized by including all eligible ICU admissions from the designated time periods. We acknowledge the possibility of limited power for subgroup analyses involving low-frequency exposures (e.g., rare ECG findings).

### 2.3. Variables

#### 2.3.1. Dependent Variable

The dependent variable is the ICU mortality (Yes/No), extracted from medical charts as the outcome of the final ICU stay.

#### 2.3.2. Independent Variables

Independent variables were grouped into seven categories: patient demographics and general information, clinical history and disease characteristics, transfusion history, vital signs, clinical symptoms, laboratory parameters, and instrumental test results. Details of the variables within each category are presented in Table 1.

### 2.4. Statistical Analysis Plan

All statistical analyses were performed using R software (version 4.3.2, released 31 October 2023). A two-sided *p*-value < 0.05 was considered statistically significant. Descriptive statistics were calculated to summarize patient demographics, clinical history and disease characteristics, transfusion history, vital signs, clinical symptoms, laboratory parameters and instrumental test results, stratified by PEWS use (Yes vs. No). Continuous variables were expressed as means with standard deviations (SD), while categorical variables were summarized using counts and percentages. Group comparisons were performed using independent *t*-tests for continuous variables after checking for distribution. Pearson’s chi-squared test was used for categorical comparisons; Fisher’s exact test was applied when expected cell counts were <5. To assess associations between PEWS use and mortality, univariable logistic regression analyses were conducted across patient demographics, clinical history and disease characteristics, transfusion history, vital signs, clinical symptoms, laboratory parameters and instrumental test results variables. Variables with *p* < 0.05 in univariable models were entered into multivariable logistic regression to adjust for confounding. Odds ratios (ORs) and adjusted odds ratios (AOR) with 95% confidence intervals (CIs) were reported. Continuous predictors were checked for outliers and non-linearity. Missing data were handled using listwise deletion for variables with <5% missingness. Variables with >5% missingness were excluded from regression modeling. Multicollinearity was assessed using variance inflation factors (VIFs), and model fit was evaluated using deviance and residual diagnostics. Original data used for the analysis are presented in Appendix A.

## 3. Results

### 3.1. General Characteristics

Table 2 presents the demographic and general characteristics of the patients stratified by PEWS monitoring status. Comprehensive participant characteristics are available in Appendix A. Mean age did not differ significantly between cohorts (7.54 ± 5.07 years vs. 7.71 ± 5.23 years, *p* = 0.79). PEWS-monitored patients were significantly less likely to be female (48.4% vs. 34.6%, *p* = 0.03) and had a lower proportion of underweight status and higher proportion of normal BMI (*p* = 0.04). Ethnic distribution was similar across groups (*p* = 0.85). Although the rate of ICU readmission during the same hospitalization was comparable (*p* = 0.29), PEWS-monitored patients had markedly fewer readmissions across separate hospitalizations (4.7% vs. 20.4%, *p* < 0.001). They also experienced longer total hospital stays (51.8 ± 25.1 vs. 38.7 ± 27.4 days, *p* < 0.001), while lengths of stay before ICU entry and within ICU were not significantly different. Mortality showed a non-significant trend toward lower deaths in the PEWS cohort (26.6% for Period 1 and 15.9% for Period 2, *p* = 0.05).

### 3.2. Factors Associated with ICU Mortality

Figure 2 summarizes the results of the multivariable logistic regression analysis, identifying key predictors of ICU mortality. Among demographic and general characteristics, male sex (AOR 0.50) and longer hospital stay before discharge (AOR 0.92) were protective factors, while prolonged ICU stay (AOR 1.15) was associated with increased mortality. Patients categorized as standard risk had significantly lower odds of death (AOR 0.42) compared to those in the high-risk group. Vital sign abnormalities, such as 4 hyperthermia episodes (AOR 5.42) and elevated heart rate (AOR 1.02), were linked to higher mortality. Similarly, the presence of SIRS (AOR 4.61), cough (AOR 2.36), and cytopenia on ICU admission (AOR 2.52) were strong symptom-based predictors of poor outcomes. Laboratory markers, including higher CRP levels (AOR 1.00) and prolonged APTT (AOR 1.04), were also associated with increased mortality risk. Among the instrumental findings, bilateral pneumonia on chest X-ray AOR (7.45) and reduced left ventricular ejection fraction on echocardiography AOR (3.03) were associated with increased mortality. Detailed results of the univariable and multivariable logistic regression results are presented in Appendix A.

## 4. Discussion

This study evaluated whether routine use of the PEWS in our oncohematology wards translated into lower ICU mortality among children with ALL. Crude ICU mortality fell from 26.6% in Period 1 to 15.9% in Period 2. However, after adjustment for demographics, disease characteristics, transfusion burden, vital signs, symptoms, laboratory parameters, and instrumental findings, PEWS use was not an independent predictor of survival. Mortality was instead associated with physiologic markers of compromise, including prolonged ICU stay, ≥4 hyperthermia episodes within 24 h of ICU admission, tachycardia, SIRS, cough, cytopenia, elevated CRP, prolonged APTT, bilateral pulmonary infiltrates, and reduced left ventricular ejection fraction. Protective factors included male sex, longer total hospital stay before discharge, and standard-risk leukemia classification. These findings suggest that once ICU-level care is required, outcomes are more strongly influenced by the severity of organ dysfunction and underlying disease biology than by prior ward-based surveillance efforts.

Despite the absence of an independent association, the observed reduction in crude mortality, coupled with PEWS’s established role in triggering early escalation, underscores its continued value as a ward-based surveillance tool. Prior research has shown that PEWS can reduce in-hospital code events and mortality across diverse pediatric populations [15]. In disease-specific studies, PEWS has demonstrated prognostic accuracy for mortality among critically ill children with leukemia, though its discriminatory ability was inferior to the Pediatric Sequential Organ Failure Assessment (pSOFA) score [21]. Notably, a multicenter study across 32 resource-limited hospitals in Latin America reported a significant reduction in mortality from clinical deterioration events following PEWS implementation [22]. Complementary evidence from a single-center study in Italy found that, among oncohematology patients, PEWS reliably predicted the need for life-saving interventions [23]. Collectively, these findings support embedding PEWS into pediatric oncology workflows, particularly when paired with organ-dysfunction scores in high-acuity settings—an approach that aligns with our current findings.

Our results have important implications for triage strategies in Kazakhstan and similar resource-constrained settings. While PEWS may enhance early detection on general wards, its predictive utility diminishes once patients reach critical illness. This supports a dual-tiered safety system: retaining PEWS for early identification while integrating ICU-based organ dysfunction scoring tools to guide escalation, triage, and prognosis. Connecting these tools to clinical action pathways, such as rapid response team activation, is essential to improving patient outcomes. This approach aligns with recommendations from international pediatric critical care networks [24,25] and could be adapted into national triage protocols under Kazakhstan’s evolving pediatric care reforms.

Several protective factors identified in our cohort warrant further discussion. Although male sex was associated with reduced ICU mortality in our analysis, this contrasts with existing epidemiological data. Studies from Peru and Ecuador report higher ALL mortality rates among males compared to females (1.7 vs. 1.2 per 100,000) [26]. Similarly, a U.S.-based SEER analysis found male sex to be associated with increased early death risk in adolescents and young adults with acute leukemia (OR > 1.0) [27]. A global analysis by Huang et al. also noted that care quality was consistently higher in females, particularly in low- and middle-income regions, potentially contributing to survival disparities [28]. These discrepancies may reflect differences in ICU-level vs. population-level dynamics, timing of interventions, or biological responses to critical illness. Further research is needed to determine whether this observed ICU survival advantage in males reflects true biologic resilience or is confounded by systemic bias in care delivery.

The protective effect of standard-risk leukemia classification is consistent with the prior literature [29]. Similarly, longer hospital stays before ICU transfer may reflect a more gradual and cautious recovery, allowing for resolution of subclinical dysfunction before escalation of care. Timing of admission may also influence outcomes; for example, a study from India reported higher mortality rates among patients admitted on weekends compared to weekdays [30].

Clinically, our findings support the integration of PEWS into pediatric oncology practice but highlight the need to augment its predictive capacity through multimodal risk stratification. We recommend combining PEWS with targeted interventions, including early cardiology consultation, proactive SIRS and fever management, aggressive cytopenia treatment, imaging-guided therapy for pulmonary infiltrates, and clear escalation protocols for children with multiple hyperthermia episodes or prolonged ICU stays. High-risk subgroups—such as females or patients with high-risk cytogenetic features—may also benefit from enhanced surveillance strategies.

Several limitations should be acknowledged. First, the retrospective, single-center design limits causal inference and generalizability. Second, post-ICU outcomes were unavailable; late deaths may therefore be under-represented. Third, some exposure categories (e.g., ventricular hypertrophy, hepatomegaly) contained few events, reducing statistical power. Fourth, residual confounding by unmeasured factors—such as genetic polymorphisms, socioeconomic status, or timing of antibiotic initiation—cannot be excluded. Fifth, continuous cardiac monitoring was not available on the oncohematology wards; outside the ICU, cardiac status was assessed intermittently via manual bedside evaluation as part of routine PEWS observations. By contrast, continuous monitoring was available in the ICU. This difference in monitoring intensity may have delayed recognition of ward-level hemodynamic instability and attenuated any detectable effect of PEWS on downstream outcomes. Sixth, the inclusion criteria differed between periods: all ICU admissions were included in Period 1, whereas only ward-based transfers were eligible in Period 2. This introduces potential selection bias. No propensity score matching or balancing methods were applied, which limits comparability across periods. Finally, clinical deterioration events are likely a more sensitive and direct measure of PEWS effectiveness than mortality alone. This study did not assess PEWS’s accuracy in predicting deterioration among pediatric ALL patients—an outcome that warrants future investigation. Prospective, multicenter studies with detailed, time-stamped physiologic data are needed to validate our findings and refine early warning systems for pediatric oncology populations. Prospective multicenter studies with granular, time-stamped physiologic data are necessary to validate these findings and refine early-warning algorithms for pediatric oncology populations.

## 5. Conclusions

In this Kazakhstani cohort of children with ALL, PEWS implementation did not independently reduce ICU mortality after controlling for a comprehensive set of demographic, clinical, physiologic, laboratory, and instrumental variables. Mortality was driven by prolonged ICU stay, ≥4 hyperthermia episodes within 24 h of ICU admission, tachycardia, SIRS, cough, cytopenia, elevated CRP, prolonged APTT, bilateral pulmonary infiltrates, and reduced left-ventricular ejection fraction. Optimizing outcomes will require combining early warning tools with aggressive deterioration management, targeted cardiopulmonary interventions, and context-specific resource allocation, rather than relying solely on surveillance scores. Future efforts should include prospective, multicenter validation studies to evaluate the predictive accuracy of PEWS in immunocompromised pediatric populations and to inform the development of real-time, algorithm-driven early warning systems tailored to the needs of low- and middle-income countries.

## Figures and Tables

**Figure 1 medicina-61-02054-f001:**
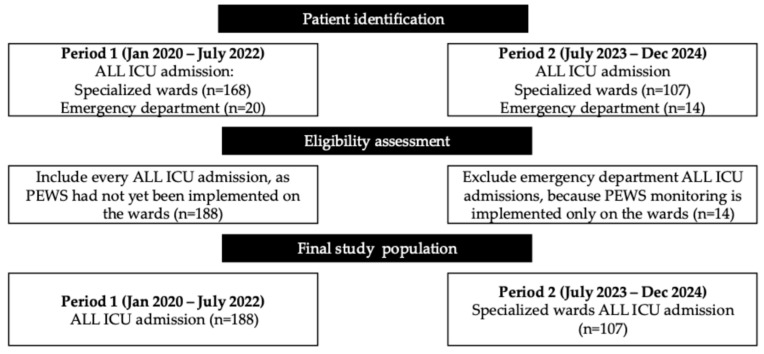
Flow diagram illustrating patient selection for the analysis of ICU admissions in children with ALL.

**Figure 2 medicina-61-02054-f002:**
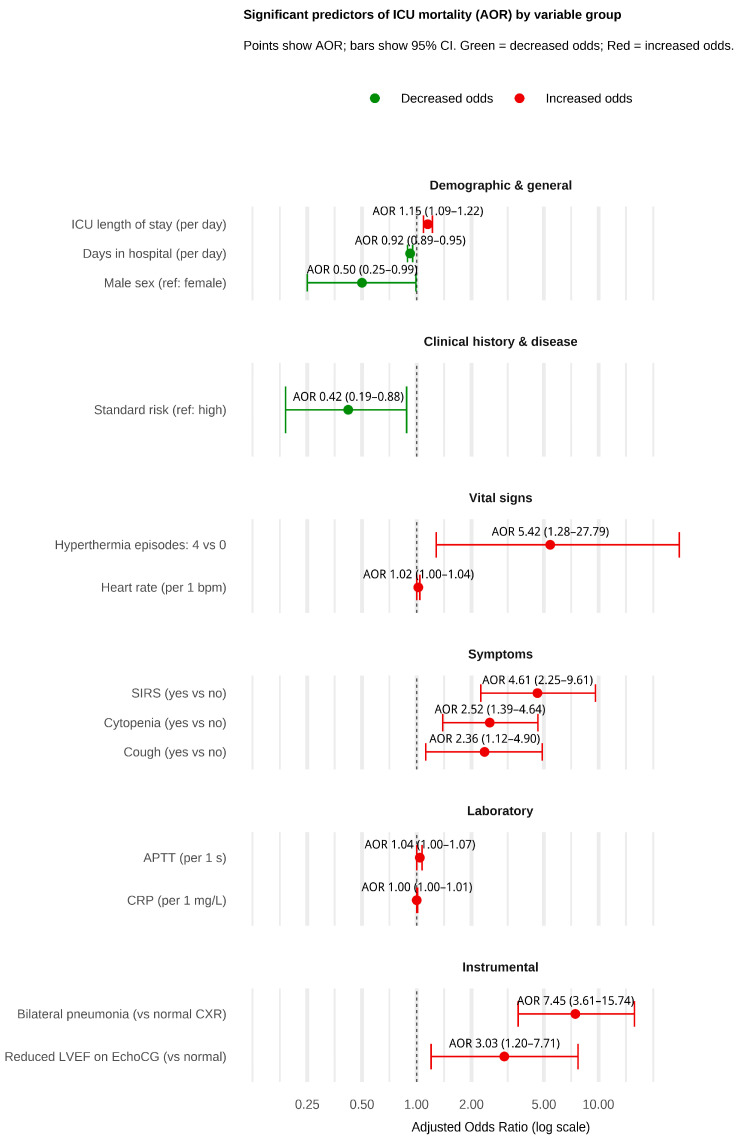
Significant predictors of ICU mortality in pediatric ALL patients.

**Table 1 medicina-61-02054-t001:** Categories of independent variables.

Category	Variables
Demographic and general	Age (years), sex (male/female), ethnicity (Kazakh/Russian/Other), BMI category (underweight/normal/overweight), number of ICU readmissions during the same hospitalization, number of ICU readmissions across different hospitalizations, total days in hospital, LoS before ICU admission, LoS in ICU and PEWS use based on the period.
Clinical history and disease characteristics	Blood group (O[I]/A[II]/B[III]/AB[IV]); Rh factor (positive/negative); ALL subtype (B-lineage/T-lineage/Other); FAB classification (L1/L2/L3); risk group (standard/intermediate/high); CNS leukemia (Yes/No); relapse (None/Once/Twice/Thrice); chemotherapy protocol (AIEOP 2009; ALL-REZ AIEOP 2009; INTERFANT; MLL-Baby AIEOP 2009-006; NHL-BFM 2004); days on chemotherapy.
Transfusion history	Total numbers of erythrocyte, platelet, plasma, and albumin transfusions.
Vital signs on ICU admission	Temperature, number of hyperthermia episodes within 24 h of ICU admission, heart rate, systolic and diastolic blood pressure, respiratory rate, saturation.
Clinical symptoms on ICU admission	SIRS, cough, hemorrhagic syndrome, and cytopenia (each Yes/No).
Laboratory parameters on ICU admission	Hemoglobin, erythrocyte count, white-cell count, absolute neutrophil count, platelet count, ESR, ALT, AST, total bilirubin, direct bilirubin, glucose, creatinine, urea, total protein, albumin, CRP, sodium, potassium, calcium, chloride, APTT, PTI, PI, INR, fibrinogen, urine specific gravity, urine pH, proteinuria.
Instrumental test results	Ultrasound: splenomegaly, hepatomegaly, lymphadenopathy, pancreatic enlargement, pleural effusion, ascitic fluid (each Yes/No). ECG: tachycardia, bradycardia, arrhythmia, normal. EchoCG: normal, pericarditis, reduced ejection fraction, ventricular hypertrophy. Chest X-ray: normal, bilateral pneumonia, right-sided pneumonia, left-sided pneumonia, lung abscess, pneumothorax with pneumonia.

Abbreviations: ALL—acute lymphoblastic leucosis; ALT—Alanine aminotransferase; APTT—Activated partial thromboplastin time; AST—Aspartate aminotransferase; BMI—body mass index; CNS—central nervous system; CRP—C-reactive protein; ECG—electrocardiogram; EchoCG—echocardiogram; ESR—Erythrocyte sedimentation rate; ICU—intensive care unit; INR—International normalized ratio; LoS—length of stay; PI—Prothrombin index; PT—Prothrombin time; Rh factor—Rhesus factor; SIRS—systemic inflammatory response syndrome.

**Table 2 medicina-61-02054-t002:** Demographic characteristics of the patients.

	PEWS Use (No) Mean ± SD/Frequency (%)	PEWS Use (Yes) Mean ± SD/Frequency (%)	*p*-Value
Demographic and General Characteristics			
Age	7.54 ± 5.07	7.71 ± 5.23	0.79
Female gender	91 (48.4)	37 (34.58)	0.03
Ethnicity			0.85
Kazakh	135 (78.95)	82 (76.64)	
Russian	10 (5.85)	8 (7.48)	
Other	26 (15.2)	17 (15.89)	
BMI			0.04
Underweight	69 (40.35)	28 (26.17)	
Normal weight	84 (49.12)	68 (63.55)	
Overweight	18 (10.53)	11 (10.28)	
ICU readmission (same hospitalization)			0.29
1	165 (87.77)	88 (82.24)	
2	20 (10.64)	15 (14.02)	
3	3 (1.60)	4 (3.74)	
ICU readmission (different hospitalizations)			<0.001
0	133 (79.64)	102 (95.33)	
1	27 (16.17)	5 (4.67)	
2	5 (2.99)	-	
3	2 (1.2)	-	
Days in hospital	38.66 ± 27.41	51.81 ± 25.12	<0.001
LoS before ICU	15.02 ± 16.37	17.39 ± 18.05	0.26
LoS in ICU	6.02 ± 7.62	6.81 ± 8	0.41
Mortality			0.05
No	138 (73.40)	90 (84.11)	
Yes	50 (26.60)	17 (15.89)	

Abbreviations: BMI—body mass index; ICU—intensive care unit; LoS—length of stay; PEWS—pediatric early warning score.

## Data Availability

The original contributions presented in this study are included in the article. Further inquiries can be directed to the corresponding author.

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
