# Peer review of "Pediatric Early Warning System (PEWS) Association with ICU Mortality in Children with Acute Lymphoblastic Leukemia: A Cohort Study from Kazakhstan"

_medicina, 2025, doi:10.3390/medicina61112054_

Round 1
Reviewer 1 Report
Comments and Suggestions for Authors
This manuscript addresses an important clinical question: the impact of PEWS monitoring on ICU mortality in children with acute lymphoblastic leukemia in Kazakhstan. It employs a retrospective cohort design and provides detailed multivariate analyses to identify independent mortality predictors. The findings are contextually relevant, particularly for resource-limited settings. However, the study has several critical methodological limitations and reporting inconsistencies that reduce its overall impact.
Title&Abstract: The title is clear and informative, appropriately reflecting the study scope, setting, and design. The abstract concisely summarizes the background, methods, key findings, and conclusion. However, instead of presenting an extensive list of statistical findings, the abstract should prioritize reporting the primary outcome and highlight only the most clinically relevant secondary result. This will enhance clarity and help readers grasp the central message more effectively.
Introduction: The manuscript provides a strong epidemiological overview of ALL globally and in Kazakhstan. It effectively highlights disparities in survival outcomes and introduces the rationale for early warning systems such as PEWS. Nonetheless, the introduction would benefit from a clearer articulation of novelty: specifically, why this study is timely and how it uniquely contributes to the current global literature. Greater emphasis could be placed on the need for rapid alert systems that are simple and easy to implement, particularly in resource-constrained settings. Additionally, it would be valuable to briefly mention ICU burden or triage inefficiencies in Kazakhstan to contextualize the clinical urgency. Since PEWS was originally developed for early sepsis detection, the authors should also clarify whether it is being used, in this context, as a predictor of overall cancer-related mortality, and discuss the potential limitations of that approach. For example, as noted in the discussion, the predictive performance of sepsis or organ dysfunction scoring systems such as SOFA could offer meaningful comparative insights.
Methods: The study appropriately categorizes covariates into seven distinct domains, and the application of multivariable logistic regression is sound. Key statistical assumptions, including multicollinearity and model diagnostics, are addressed. However, no sample size or power calculation is provided, making it unclear whether the study was adequately powered to detect a meaningful difference in mortality. Furthermore, the inclusion of patients only from wards with PEWS monitoring in the post-implementation period (Period 2), but from all ICU admissions in the pre-PEWS period (Period 1), introduces a risk of selection bias and baseline imbalances. The authors should consider performing propensity score matching or conducting sensitivity analyses to mitigate this bias. There is also no discussion of missing data handling. While the manuscript refers to the CONSORT diagram, it is important to note that CONSORT is intended for randomized controlled trials. The STROBE statement is the appropriate guideline for this type of observational, retrospective cohort study, and should be adopted accordingly. A flow diagram aligned with STROBE standards and a more formal clarification of inclusion and exclusion criteria are needed. While the detailed variable table is appreciated, given the already high number of tables, presenting this information in text form may reduce redundancy and enhance readability.
Results: The findings are reported in detail and clearly organized according to the seven domains. However, results presented in tables are repeated extensively in the text, which reduces efficiency. This is particularly evident in the lead-ins to Tables 3–8. The authors should avoid duplicating full results within the narrative and instead focus on interpreting the most clinically meaningful outcomes. Table 2 is overly long and may benefit from being split. More broadly, the entire Results section is dense and contains excessive secondary analysis relative to the central research question. The Results should be redesigned around the primary outcome, with peripheral analyses moved to Supplementary Material where appropriate. Additionally, the presentation would benefit from forest plots, which would reduce the number and length of tables while visually summarizing adjusted odds ratios for key variables. PEWS is a score, yet the manuscript does not evaluate its discriminatory performance. It would strengthen the findings significantly to include receiver operating characteristic (ROC) curves, area under the curve (AUC), and analysis of sensitivity and specificity at various PEWS thresholds. If these are not available, an explanation should be provided.
Discussion: The discussion is clinically thoughtful and provides a well-informed interpretation of the study’s findings. The authors compare their results to international literature, including both high- and low-resource settings. The proposal to combine PEWS with organ dysfunction scores is logical and consistent with their findings. Accompanied by the authors' findings, greater emphasis should be placed on how these results can inform national triage protocols and early-warning frameworks in similar resource-constrained countries. The authors should clearly delineate what their study demonstrates versus what it suggests or recommends. Additionally, it is important to discuss whether the PEWS thresholds were optimized or appropriate for this specific population, as this could significantly influence outcome prediction.
Conclusion: The conclusion is well-aligned with the findings and appropriately cautious. It acknowledges that PEWS alone did not independently reduce ICU mortality and correctly emphasizes the need for combined strategies, including targeted cardiopulmonary management and resource allocation. The authors may consider briefly suggesting future directions, such as prospective validation studies or real-time algorithm-based PEWS optimization.
Language&Formatting: The manuscript contains minor grammatical and formatting issues, including missing hyphens or spacing (e.g., “highacuity,” “nonPEWS”). These should be corrected. The abstract contains typographical inconsistencies (e.g., “26.6%; OR”), which should be fixed for clarity.
Author Response
Dear Reviewer,
Thank you very much for your constructive and thoughtful feedback.
Please find our detailed responses attached.
Response to the Reviewer 1:
Title&Abstract: The title is clear and informative, appropriately reflecting the study scope, setting, and design. The abstract concisely summarizes the background, methods, key findings, and conclusion. However, instead of presenting an extensive list of statistical findings, the abstract should prioritize reporting the primary outcome and highlight only the most clinically relevant secondary result. This will enhance clarity and help readers grasp the central message more effectively.
Authors Response: Thank you for this helpful suggestion. We revised the Abstract to foreground the primary outcome and present only clinically salient secondary outcomes as highlighted in the revised text.
____________________________________________________________________________
Introduction: The manuscript provides a strong epidemiological overview of ALL globally and in Kazakhstan. It effectively highlights disparities in survival outcomes and introduces the rationale for early warning systems such as PEWS. Nonetheless, the introduction would benefit from a clearer articulation of novelty: specifically, why this study is timely and how it uniquely contributes to the current global literature. Greater emphasis could be placed on the need for rapid alert systems that are simple and easy to implement, particularly in resource-constrained settings. Additionally, it would be valuable to briefly mention ICU burden or triage inefficiencies in Kazakhstan to contextualize the clinical urgency. Since PEWS was originally developed for early sepsis detection, the authors should also clarify whether it is being used, in this context, as a predictor of overall cancer-related mortality, and discuss the potential limitations of that approach. For example, as noted in the discussion, the predictive performance of sepsis or organ dysfunction scoring systems such as SOFA could offer meaningful comparative insights.
Authors Response: We thank the reviewer for the thoughtful and constructive feedback. In response, we have revised the introduction to more clearly articulate the novelty and timeliness of our study, emphasizing the lack of prior data from Kazakhstan regarding the use of PEWS in pediatric oncology, as suggested. All changes are highlighted in the text, with additional references.
____________________________________________________________________________
Methods: The study appropriately categorizes covariates into seven distinct domains, and the application of multivariable logistic regression is sound. Key statistical assumptions, including multicollinearity and model diagnostics, are addressed. However, no sample size or power calculation is provided, making it unclear whether the study was adequately powered to detect a meaningful difference in mortality. Furthermore, the inclusion of patients only from wards with PEWS monitoring in the post-implementation period (Period 2), but from all ICU admissions in the pre-PEWS period (Period 1), introduces a risk of selection bias and baseline imbalances. The authors should consider performing propensity score matching or conducting sensitivity analyses to mitigate this bias. There is also no discussion of missing data handling. While the manuscript refers to the CONSORT diagram, it is important to note that CONSORT is intended for randomized controlled trials. The STROBE statement is the appropriate guideline for this type of observational, retrospective cohort study, and should be adopted accordingly. A flow diagram aligned with STROBE standards and a more formal clarification of inclusion and exclusion criteria are needed. While the detailed variable table is appreciated, given the already high number of tables, presenting this information in text form may reduce redundancy and enhance readability.
Authors Response:
We thank the reviewer for the insightful comments, which helped us improve the methodological clarity and transparency of the manuscript. In response:
- We have added a rationale in Section 2.2 acknowledging the absence of a priori power calculation, justified by the retrospective design and inclusion of all eligible ICU admissions. We also noted the potential for reduced power in rare event subgroups.
- We agree that differing inclusion criteria between Period 1 and Period 2 could introduce bias. To address this, we added the potential for selection bias to the limitations section of the manuscript.
- A formal approach to missing data is now described, including listwise deletion for variables with <5% missingness and exclusion of variables with higher missingness.
- Figure 1: We acknowledge the previous misattribution of CONSORT. We have deleted the name CONSORT flow diagram from the title of the Figure 1.
- Table 1: We have moved the extensive Tables from the results section of the manuscript to the supplementary materials. Therefore, Table 1 is retained in the manuscript to provide the full description of study variables.
We hope these changes adequately address the reviewer’s concerns and strengthen the methodological rigor of the manuscript.
____________________________________________________________________________
Results: The findings are reported in detail and clearly organized according to the seven domains. However, results presented in tables are repeated extensively in the text, which reduces efficiency. This is particularly evident in the lead-ins to Tables 3–8. The authors should avoid duplicating full results within the narrative and instead focus on interpreting the most clinically meaningful outcomes. Table 2 is overly long and may benefit from being split. More broadly, the entire Results section is dense and contains excessive secondary analysis relative to the central research question. The Results should be redesigned around the primary outcome, with peripheral analyses moved to Supplementary Material where appropriate. Additionally, the presentation would benefit from forest plots, which would reduce the number and length of tables while visually summarizing adjusted odds ratios for key variables. PEWS is a score, yet the manuscript does not evaluate its discriminatory performance. It would strengthen the findings significantly to include receiver operating characteristic (ROC) curves, area under the curve (AUC), and analysis of sensitivity and specificity at various PEWS thresholds. If these are not available, an explanation should be provided.
Authors Response: The authors sincerely thank the reviewer for their thorough and constructive feedback. We fully agree with the recommendations to improve clarity, focus, and presentation within the Results section. The following revisions have been made accordingly:
- To reduce redundancy and improve readability, we have removed detailed regression results from the main text. These tables (previously Tables 3–8) are now included in the Supplementary Material (Tables S2–S7).
- We have split Table 2 as suggested. Only the demographic and general characteristics are retained in the main manuscript (revised Table 2), while other domains are relocated to the Supplementary Material (Table S1).
- We have added a forest plot to visually summarize the adjusted odds ratios (AORs) for the key predictors of ICU mortality, including PEWS use and other significant covariates (Figure 2). The Results section text has been revised to focus on the main findings of the study.
- We agree that assessing the discriminatory performance of PEWS using ROC curves, AUC, and sensitivity/specificity analysis would add significant value. However, such analyses were not feasible in this study due to the absence of complete PEWS score documentation across all patients on the onco-hematology wards before transfer to the ICU. We have added a clear explanation on the PEWS use in the introduction section, and o acknowledge the absence of the ROC analysis, we have added the following statement to the Limitations section
____________________________________________________________________________
Discussion: The discussion is clinically thoughtful and provides a well-informed interpretation of the study’s findings. The authors compare their results to international literature, including both high- and low-resource settings. The proposal to combine PEWS with organ dysfunction scores is logical and consistent with their findings. Accompanied by the authors' findings, greater emphasis should be placed on how these results can inform national triage protocols and early-warning frameworks in similar resource-constrained countries. The authors should clearly delineate what their study demonstrates versus what it suggests or recommends. Additionally, it is important to discuss whether the PEWS thresholds were optimized or appropriate for this specific population, as this could significantly influence outcome prediction.
Authors Response: We sincerely thank the reviewer for their thoughtful evaluation of the Discussion section. In response, we have substantially revised the Discussion to differentiate between what our study demonstrates (e.g., physiologic predictors of ICU mortality) and what it suggests or recommends (e.g., integration of PEWS into dual-tier surveillance strategies). We also now emphasize how the findings can inform national triage frameworks in Kazakhstan and similar low- and middle-income settings. We clarified that PEWS was not evaluated for optimal cut-off thresholds or discriminative accuracy, due to the absence of complete, numeric PEWS data for all patients. This limitation has been added explicitly to the final paragraph of the manuscript. We also discussed the role of PEWS in combination with organ dysfunction scores, consistent with both our results and global best practices with appropriate references. We hope these refinements improve the clinical relevance and scientific clarity of our interpretation. Thank you again for your valuable feedback.
____________________________________________________________________________
Conclusion: The conclusion is well-aligned with the findings and appropriately cautious. It acknowledges that PEWS alone did not independently reduce ICU mortality and correctly emphasizes the need for combined strategies, including targeted cardiopulmonary management and resource allocation. The authors may consider briefly suggesting future directions, such as prospective validation studies or real-time algorithm-based PEWS optimization.
Authors response: We thank the reviewer for their positive assessment of the Conclusion and appreciate the valuable suggestion to outline future research directions. In response, we have revised the Conclusion to include specific recommendations for prospective validation studies and real-time algorithmic optimization of PEWS in pediatric oncology populations.
____________________________________________________________________________
Language&Formatting: The manuscript contains minor grammatical and formatting issues, including missing hyphens or spacing (e.g., “highacuity,” “nonPEWS”). These should be corrected. The abstract contains typographical inconsistencies (e.g., “26.6%; OR”), which should be fixed for clarity.
Authors Response: We thank the reviewer for noting these issues. The manuscript has been thoroughly revised by a professional English language editor, and all grammatical errors, typographical inconsistencies, and formatting issues—including missing hyphens and spacing—have been corrected throughout the text and abstract to improve clarity and readability.
Reviewer 2 Report
Comments and Suggestions for Authors
The authors present a retrospective cohort study about PEWS in order to detect the predictor factors for mortality and as a result to diminish the mortality rate in future.
The article is dis-organized and tables are not explanatory. The tables in an article should be concise model of what an authors describe in the text. However, the authors in this article put all demographic and laboratory values in tables. They should insert only main findings into tables. And the number of tables shouldn’t exceed 5 (five).
The other point is statistical analysis. An expert should do analysis with adjustments since length of days in hospital was found a protector factor!
Another point is lack of language editing. A language editing is mandatory.
The discussion section is main part of article. However, I found almost 2 paragraph for discussion despite many findings authors put forward. The discussion section should be expanded
Here is my detailed report as following:
Line 42: Please describe non-PEWS
Line 44-51: Please give p value for each of AOR
Line 51-52: Please use clearer sentence
Line 61: “Leukemia is a malignancy that arises in the bone marrow and involves the body’s 61
white blood cells” This sentence should not be used as an introductory sentence in a highly respected journal. Please write a more serious sentence.
Line 84: The sentence is not related to Citation 10, which you used in your article. Please give more appropriate citation for that sentence.
Page 6, Table 2: What did you mean by classifying the risk groups of ALL as high, medium, standard? Is there any medium risk group? Or did you mean intermediate?
Page 6 Table 2: Please use “chemotherapy protocols” instead of chemo type
Page 7 Table 2: What did you mean by writing hyperthermia times? Please clarify
Page 7 Table 2: What kind of hemorrhage did patients have? Please clarify
Page 7 Table 2: What does the number next to the laboratory values mean? For i.e hemoglobin 89.69 (18.88)?
Comments on the Quality of English Language
Another point is lack of language editing. A language editing is mandatory.
Author Response
Dear Reviewer,
Thank you very much for your constructive and thoughtful feedback.
Please find our detailed responses attached.
The authors present a retrospective cohort study about PEWS in order to detect the predictor factors for mortality and as a result to diminish the mortality rate in future.
The article is dis-organized and tables are not explanatory. The tables in an article should be concise model of what an authors describe in the text. However, the authors in this article put all demographic and laboratory values in tables. They should insert only main findings into tables. And the number of tables shouldn’t exceed 5 (five). The other point is statistical analysis. An expert should do analysis with adjustments since length of days in hospital was found a protector factor!
Authors response: The authors acknowledge the reviewer comment and agree with the reviewer. We now present the forest plot with significant findings of the present study, and all tables with logistic regression results have been moved to the Supplementary materials. Our analysis was evaluated by the statistician, and we have performed required adjustment to avoid Type I and Type II errors.
Another point is lack of language editing. A language editing is mandatory.
Authors response: The authors acknowledge the reviewer comment and agree with the reviewer. The manuscript has been extensively revised to improve the grammar and readability of the manuscript.
The discussion section is main part of article. However, I found almost 2 paragraph for discussion despite many findings authors put forward. The discussion section should be expanded
Authors response: The authors acknowledge the reviewer comment and agree with the reviewer. The discussion section has been revised and expanded to compare and discuss the findings of the present analysis. All changes are highlighted in the text.
Here is my detailed report as following:
Line 42: Please describe non-PEWS
Authors response: The authors acknowledge the reviewer comment and agree with the reviewer. Non-PEWS was revised to prior to PEWS implementation.
Line 44-51: Please give p value for each of AOR
Authors response: The authors acknowledge the reviewer comment and agree with the reviewer. P-values have been removed from the abstract
Line 51-52: Please use clearer sentence
Authors response: The authors acknowledge the reviewer comment and agree with the reviewer. The conclusion of the abstract has been revised for clarity.
Line 61: “Leukemia is a malignancy that arises in the bone marrow and involves the body’s 61
white blood cells” This sentence should not be used as an introductory sentence in a highly respected journal. Please write a more serious sentence.
Authors response: The authors acknowledge the reviewer comment and agree with the reviewer. The sentence was deleted from the introduction section
Line 84: The sentence is not related to Citation 10, which you used in your article. Please give more appropriate citation for that sentence.
Authors response: The authors acknowledge the reviewer comment. The references for the provided statement have been revised as follows:
- Kembhavi, S.A.; Somvanshi, S.; Banavali, S.; Kurkure, P.; Arora, B. Pictorial Essay: Acute Neurological Complications in Children with Acute Lymphoblastic Leukemia. Indian J Radiol Imaging 2012, 22, 98, doi:10.4103/0971-3026.101080.
- Macaluso, A.; Genova, S.; Maringhini, S.; Coffaro, G.; Ziino, O.; D’Angelo, P. Acute Respiratory Distress Syndrome Associated with Tumor Lysis Syndrome in a Child with Acute Lymphoblastic Leukemia. Pediatr Rep 2015, 7, 5760, doi:10.4081/PR.2015.5760.
Page 6, Table 2: What did you mean by classifying the risk groups of ALL as high, medium, standard? Is there any medium risk group? Or did you mean intermediate?
Authors response: The authors acknowledge the reviewer comment. As we have specified in Table 1 the intendent and proper description is intermediate risk. The required changes have been implemented in Tables and the Figure 1.
Page 6 Table 2: Please use “chemotherapy protocols” instead of chemo type
Authors response: The authors acknowledge the reviewer comment and agree with the reviewer. The chemo type has been revised to chemotherapy protocols. All changes are highlighted.
Page 7 Table 2: What did you mean by writing hyperthermia times? Please clarify
Authors response: The authors thank the reviewer for the insightful comment and offer the following clarification: As specified in Table 1, the term refers to hyperthermia episodes occurring within 24 hours of ICU admission. We have revised the manuscript to use this term consistently throughout the text.
Page 7 Table 2: What kind of hemorrhage did patients have? Please clarify
Authors response: The authors thank the reviewer for the insightful comment and offer the following clarification: The term refers to hemorrhagic syndrome. We have revised the manuscript to use this term consistently throughout the text.
Page 7 Table 2: What does the number next to the laboratory values mean? For i.e hemoglobin 89.69 (18.88)?
Authors response: The authors thank the reviewer for the insightful comment and offer the following clarification: The values mean Mean (SD) / Frequency (%). To avoid the confusion (SD) values have been. Replaced with ±. Now the column headers read PEWS use (No) Mean±SD / Frequency (%) PEWS use (Yes) Mean±SD / Frequency (%)
Reviewer 3 Report
Comments and Suggestions for Authors
See attached

Author Response
Dear Reviewer,
Thank you very much for your constructive and thoughtful feedback.
Please find our detailed responses attached.
Review of “PEWS Association with ICU Mortality in Children with ALL”
COMMENTS AND QUESTIONS
Period 1 111 (January 2020 – December 2022): PEWS monitoring had not yet been implemented; there- 112 fore, every ICU admission (from onco- hematological wards and emergency department) 113 of children with ALL was included. Period 2 (September 2023 – December 2024): after full 114 implementation of PEWS monitoring in the four specialized onco- hematology wards, 115 only patients transferred from these wards were eligible.
It seems that outcome improvement may have been mostly related to specialized hemonc ICU with trained nursing staff. This is not to discount your medical data comparison study which is rigorous and well done. Can you comment?
Authors response: Thank you for your thoughtful comment. We acknowledge that the PEWS implementation is linked to the specialized ICU nursing staff training in Period 2. However, the introduction of PEWS monitoring itself plays a critical role in early detection and timely escalation of care, which we believe contributed significantly to the observed patient outcomes.
For patients with multiple ICU stays during a single hospitalization, only data 118 from the final ICU admission were analyzed. For patients with several hospitalizations, 119 the last hospitalization was used.
Please report on the number of prior hospitalizations per patient. Perhaps a larger number would be significant in univariate analysis of outcome.
Authors response: Thank you for your insightful comment. We fully understand the importance of considering prior hospitalizations as a potential variable influencing patient outcomes. However, we were unable to include this data to our analysis, as patients may have been hospitalized across different healthcare organizations, including both public and private hospitals. Unfortunately, linking patient data across various institutions is not straightforward. Additionally, as our study is based on a retrospective review of hospital records, we only had access to data from the hospital where the analysis was conducted. Therefore, we could not reliably collect information on prior hospitalizations from other institutions where patients may have been treated.
Regarding the multiple significant factors in the univariate analysis (UVA) some are likely to be not significant do to the problem of multiple analyses. This is would be likely as many of the borderline significance in UVA were not in multivariate analysis.
Authors response: The authors acknowledge the reviewer comment. We agree that the issue of multiple comparisons could have influenced the results of the univariate analysis (UVA). Only significant variables were included to the MVA, which we believe reflects the adjustment for potential confounders and the reduction in Type I errors. We have carefully considered this issue in our analysis to ensure the robustness of our findings.
PEWS implementation retained significance, 232 conferring roughly a 48% reduction in mortality (AOR 0.52, 95% CI 0.27–0.98; p = 0.05) 233 compared to the period when it was not implemented
Also you have a p value of 0.05 (not < 0.05) to be significant. You should say significant due to the odds ratio confidence interval not containing 1 not due to p value which should be < 0.05 if not rounded.
Authors response: The authors acknowledge the reviewer’s thoughtful comments and agree with the reviewer. The PEWS values and interpretation in the text were revised. To avoid multiple testing, and avoid Type I error, we tested the PEWS implementation only in the first model, and statistical significance was set to <0.05.
MINOR SUGGESTIONS AND CORRECTIONS
“Neuroleucosis” -Change to CNS Leukemia. Its a nice Latin term but not well known. I had to look it up as I had never heard of it
Authors response: The authors acknowledge the reviewer’s thoughtful comments and agree with the reviewer. We revised the neuroleucosis to CNS leukemia.
Table 2
The p-value of SIRS <0.001 seems not to match the patient distributions which are close. Please check
Authors response: The authors acknowledge the reviewer’s comments. SIRS is a significant predictor of ICU mortality. The presented difference is correct.
The AST p-value is missing trailing digits. 0.00 Same for PTI
Authors response: The authors acknowledge the reviewer’s comments. 0.00 values have been corrected to <0.001 throughout the manuscript.
Male sex was associated with a 50% 187 reduction in the odds of death (AOR 0.50; 95% CI 0.25–0.99; p = 0.04) compared with fe- 188 male sex.
I am wondering if there are other reasons for females having a higher risk of death then males. As we know females have better survival in many conditions that men. Is it possible that it is gender bias and not gender. Male female gender bias has been part of human existence since the first humans. If this is a sensitive subject for you we can let
it go. I have three abstracts for you to review specifically for ALL including one in Peru.
Three Abstracts showing male pediatric ALL patients have higher death in remission than females.
Authors response: The authors acknowledge the reviewer’s thoughtful comments and agree with the reviewer that this issue warrants a further discussion, and is a complicated topic for discussion. We have revised the manuscript to discuss the protective factors and the role of observed gender differences. All changes are highlighted in the text.